# The Earliest Corotocini (Insecta: Coleoptera: Staphylinidae) from Dominican Amber, with Remarks on Post-Imaginal Growth Influence on Termitophile Taxonomy [note 1]

**DOI:** 10.3390/insects13070614

**Published:** 2022-07-08

**Authors:** Bruno Zilberman, Zi-Wei Yin, Chen-Yang Cai

**Affiliations:** 1Museu de Zoologida, Universidade de São Paulo, Caixa Postal 42494, São Paulo 04218-970, Brazil; brunozilberman@usp.br; 2College of Life Sciences, Shanghai Normal University, Shanghai 200234, China; pselaphinae@gmail.com; 3State Key Laboratory of Palaeobiology and Stratigraphy, Nanjing Institute of Geology and Palaeontology, and Center for Excellence in Life and Paleoenvironment, Chinese Academy of Sciences, Nanjing 210008, China; 4School of Earth Sciences, University of Bristol, Life Sciences Building, Tyndall Avenue, Bristol BS8 1TQ, UK

**Keywords:** Miocene, physogastry, post-imaginal growth, stenogastry, termitophily

## Abstract

**Simple Summary:**

The tribe Corotocini is one of the most diverse and morphologically specialized groups of rove beetles associated with termites. All species of Corotocini present some degree of membranous enlargement of the abdomen, called physogastry. The development of physogastry occurs after the beetle emerges from the pupa as an ordinary beetle and is accompanied by further modifications related to its sclerotized parts, which makes the early stages (stenogastrics) strikingly different from the fully developed forms (physogastrics). The present study reports the very first fossil record for Corotocini, with a new genus and species, *Pareburniola dominicana* gen. et. sp. nov., from Miocene Dominican Republic amber. The fossil is represented by a stenogastric individual, which poses challenges in the taxa description since the last stage of development remains unknown. We utilize the current knowledge of post-imaginal growth in Corotocini to understand what is likely and not likely to change in the morphology during this phenomenon.

**Abstract:**

*Pareburniola dominicana* Zilberman, Yin & Cai gen. et. sp. nov. is the very first fossil record of the tribe Corotocini, reported from Miocene Dominican Republic amber. The new species, which is based on a stenogastric individual, is described and illustrated and is included in the subtribe Corotocina due to the combination of a tarsal formula 4-4-4, an elongated gula, a developed labial palp, a reduced fourth palpomere, separated metacoxae and a glandular structure on the posterior region of the head. Since the taxon belongs to the physogastric tribe Corotocini, which presents post-imaginal growth, this phenomenon is herein discussed, and its current knowledge is used to understand the possible outcomes during morphological changes in the fossil species.

## 1. Introduction

Dominican amber is one of the largest occurrences of Miocene fossil resin in the world and contains the most comprehensive number of insect fossils from a single area in the Americas [1,2]. Along with many important insect fossils that contribute significantly to different areas of entomological knowledge, Dominican amber now has the distinction of providing the first fossil record for the remarkable termitophilous tribe Corotocini.

The tribe Corotocini is the most successful exclusively termitophilous lineage, consisting primarily of pantropical rove beetles with more than 270 species. It is included in the subfamily Aleocharinae, seemingly presenting evolvability towards social parasitism [3]. Corotocine beetles all show some degree of physogastry and usually have mentum fused to submentum, unmargined mesocoxal acetabula, and terminal antennomeres with two or more coeloconic sensilla [4]. In recent years, there has been increasing interest in studying this tribe, with ongoing master’s and PhD projects, and many works have already been published in various areas [5,6,7,8,9,10,11,12,13,14].

Other fossils of termitophilous rove beetles have already been recorded, including cretaceous trichopseniine taxa, which shed light on the origin of termitophily [15,16]. There is even a record, also of trichopseniine, in Dominican amber [17]. However, this is the first time that a physogastric higher-termite-associated tribe of beetles has been found in the fossil record. *Pareburniola dominicana* gen. et sp. nov. is herein described, and its importance and biogeographical significance are discussed in the light of modern termite systematics. Because the tribe presents post-imaginal growth and because the fossil is represented by a stenogastric individual, some inferences should be made with caution, which is also discussed in detail.

## 2. Materials and Methods

The holotype studied here is housed in the Nanjing Institute of Geology and Palaeontology (NIGP), Chinese Academy of Sciences, Nanjing, China. The amber piece was excavated from the “Los Brachos” (La Cumbre) mine in Cordillera Septentrional (Santiago). The age, origin, and faunal diversity of Dominican amber have been recently summarized [18,19,20]. The photograph under incident light (Figure 1) was taken using a Fujifilm X-T2 camera equipped with a Nikon PB-6 Bellows (Tokyo, Japan) and a Nikon 19 mm f/2.8 Macro-Nikkor lens (Tokyo, Japan) with a Fostec EKE DCR II LR92240 20,500 Fiber Optic Light Source (FOSTEC, Ansan, Korea) as the light source. Helicon Focus v. 8.1.1 and Zerene Stacker v. 1.04 were used for image stacking. Using the same method as Li et al. [21], the amber inclusion was imaged using high-resolution X-ray microtomography (micro-CT) to uncover fine morphological detail. Scans were made using a Zeiss Xradia 520 versa at the micro-CT laboratory of the Nanjing Institute of Geology and Palaeontology, CAS. A CCD-based 4× objective was used, providing isotropic voxel sizes of 1.8775 μm with the help of geometric magnification. During the scanning, the acceleration voltage of the X-ray source was 40 kV. To improve the signal-to-noise ratio, 3001 projections over 360° were collected, and the exposure time for each projection was 6 s. The tomographic data were analyzed and visualized using Avizo ver. 2019.1 with an Isosurface analog (Figure 2) and a volume rendering method (Figure 3A,B). Additional photographs of the Corotocina, *Corotoca pseudomelantho* Zilberman, 2020 and *Thyreoxenus solomonensis* Seevers, 1937 were taken using a Zeiss LEO 440 for scanning electron microscopy (SEM) with the specimen covered with gold (Figure 3C,D) and a Zeiss Axioscope with a camera attached for light microscopy of the slide mounts. *Corotoca pseudomelantho* specimens are housed in the Museu de Zoologia da Universidade de São Paulo (MZSP) (MZSP 27309, Figure 3C–E, MZSP 27306), and the *T*. *solomonensis* slide was a loan from the Field Museum of Natural History (FMNH) (Figure 3F, No. 19173). All images were further processed and arranged onto plates in Adobe Photoshop CC ver. 20.0.1.

## 3. Results

### Systematic Paleontology

Order Coleoptera Linnaeus, 1758

Family Staphylinidae Latreille, 1802

Subfamily Aleocharinae Fleming, 1821

Tribe Corotocini Fenyes, 1918

Subtribe Corotocina Fenyes, 1918


***Pareburniola* Zilberman, Yin & Cai, gen. nov.**


**Type species:** *Pareburniola dominicana* sp. nov.

**Diagnosis:** *Male*. Unknown. *Female* (Figure 1, Figure 2 and Figure 3A,B). The specimen is small, measuring about 1.4 mm from the head to the abdominal apex (stenogastric); the antennomeres are transverse to the subquadrate, with antennomeres 2 and 10 shorter than any other; the eyes are large; the head has a longitudinal suture on the vertex; the endosternite lateral projections are short; the first tarsomeres of the meso- and metalegs are long, about the same length as the subsequent tarsomeres combined; there are rows of long setae on the sternites; tergite X is divided into two lobes and bears very long and bicolored setae.

**Comparative notes:***Pareburniola* morphologically resembles the extant Neotropical genus *Eburniola* Mann in sharing transverse to subquadrate antennomeres, a strong longitudinal suture on the vertex, a reduced fourth maxillary palp, a metacoxae that is narrowly separated by shorter metendosternite lateral projections, rows of long setae on sternites III–VII, and tergite X divided into two lobes and bearing very long and bicolored setae; however, it clearly differs from the latter with bigger eyes, narrower gula, antennomeres that are not telescoped and with visible pedicels between them, antennomere 10 shorter than other antennomeres except antennomere 2, an elytron longer than it is wide, posterior coxae that are longer than they are wide, and long first tarsomeres of the meso- and metalegs. *Pareburniola* also resembles the Afrotropical genera *Termitomimus* Trägårdh and *Nasutimimus* Kistner, especially in the head capsule and thorax, which are quite similar to those of *Eburniola*. However, these Afrotropical genera have vital differences in the abdomen, bearing much shorter setae on the sternites, with tergite X being represented by a single subquadrate piece as well as bearing short setae. It is noteworthy that *Termitomimus* and *Nasutimimus* also have a reduced fourth maxillary palp but not to a point where it is indistinct.

**Etymology:** The generic name *Pareburniola* is a junction of Para- (ancient Greek, “παρά”; near, next to), and the extant genus name, *Eburniola*, to which the fossil is similar. The specific name *dominicana* is derived from the locality, the Dominican Republic, and the fossil resin, Dominican amber. The gender is feminine.


***Pareburniola dominicana* Zilberman, Yin & Cai, sp. nov.**


**Type material:** NIGP180476; a complete, well-preserved female entombed in an approximately 11 mm × 6 mm Dominican amber piece; deposited in NIGP.

**Diagnosis: Same as** for the genus.

**Description:** The head is slightly longer than it is wide and is the widest before the eyes; the eyes are large and occupy a great portion of the lateral region of the head; the gula is short and widens slightly by less than half of the total head width through the posterior region; the vertex has a strong and longitudinal suture in the midline, covering about the whole dorsal region of the head; the glandular structure is present on the posterior region of the head (Figure 2A–C); the antenna have 11 antennomeres: the scape is elongated to about the same length of antennomeres 2–4 and combined with a small portion of antennomere 5; antennomeres 2–9 are transverse to subquadrate, with antennomere 2 shorter than any other, antennomeres 3–9 about the same length, and antennomere 10 shorter than the previous antennomeres, with the exception of antennomere 2; antennomere 11 is about the same length as antennomeres 9–10 combined (Figure 1); the maxilla has three visible palpomeres; the fourth palpomere is completely reduced (Figure 1 and Figure 3A, B); the first palpomere is small and subquadrate; the second palpomere is slightly longer than it is wide; the third palpomere is very large, elongated, and longer than the second palpomere; the prementum has a broad ligula that is rounded in the lateral margins, pronounced, and bilobed apically; the labial palp is developed; the mentum is fused to the submentum (postmentum) (Figure 2C,D).

Thorax: The pronotum is wider than it is long and is not apparently impressed, and the anterior margin is elevated (Figure 1 and Figure 2E); the elytron is longer than it is wide (Figure 2E); the hind wings are present (Figure 1); the mesosternum is shorter than the metasternum. The legs are well-developed: pro-meso- and metacoxae are elongated; the metacoxae are distinctly shorter and wider and narrowly separated (Figure 2G,H); the tarsal formula 4-4-4 has the first tarsomere of pro-meta legs that are long—about the same length of the subsequent tarsomeres combined (Figure 1).

The abdomen is stenogastric, with sternites that are broader than tergites, and it is strap-like; sternites III–VII each bear a row of long setae; tergites III–VII also have setae, but they are shorter; tergite X is divided into two elongated portions, bearing both long and bicolored setae.

**Etymology:** The specific epithet refers to the Dominican Republic, where the fossil originated.

## 4. Discussion

A tricky step in the placement of the fossil is separating the subtribe Corotocina from Termitoptochina, as the latter share the tarsal formula 4-4-4, reduced mouthparts in all species, and the head gland in two genera with the former. Another characteristic shared by both is the reduction in the fourth palpomere in some groups. It occurs in *Cavifronexus* Zilberman and *Corotoca* Schiødte, *Eburniola* Mann (at least in *E*. *lujae* Seevers, 1957 and *E*. *gastrovittata* Seevers, 1937) and to some extent in *Nasutimimus* and *Termitomimus* (Corotocina); it is also reduced in *Hospitaliptochus* Jacobson, Kistner & Pasteels, *Lacessiptochus* Jacobson, Kistner & Pasteels and *Paracorotoca* Warren (Termitoptochina). According to published literature, the Corotocina and Termitoptochina genera have filamentous sensillae on a reduced fourth palpomere, and the basal part of the sensillae are often confused with the palp (Figure 3C–E) [9,22]. Notably, these filamentous sensillae are also present in groups with a developed fourth maxillary palp (Figure 3F). The complete reduction om the fourth maxillary palp, which leaves only the filamentous sensillae visible, is seemingly a condition present in *Pareburniola* (Figure 3A,B). However, the subtribe Termitoptochina can be ruled out, as the gula is quite distinct, much shorter than in most Corotocina, widening through the apex and occupying a significant part of the head width; additionally, the labial palp is extremely reduced in most genera when compared to those of Corotocina, and the metacoxae are usually not separated by lateral extensions of the metendosternite [22].

Therefore, *Pareburniola* is included in the subtribe Corotocina by the combination of a tarsal formula 4-4-4, a reduced fourth maxillary palp, a developed labial palp, an elongated gula, a glandular structure on the posterior region of the head, and metacoxae that are separated by a metastesternal process. Furthermore, the transverse to subquadrate antennomeres, strong longitudinal suture on the vertex, seemingly shorter metendosternite lateral projections, rows of long setae on sternites III–VII, and tergite X being divided into two lobes and bearing very long and bicolored setae places the genus close to the Neotropical *Eburniola*, from which it can be distinguished by its bigger eyes, narrower gula, antennomeres that are not telescoped and that have visible pedicels between them, an antennomere 10 shorter than the other antennomeres except antennomere 2, an elytron that is longer than it is wide, posterior coxae longer than they are wide, and long first tarsomeres of the meso- and metalegs.

### 4.1. Stenogastry, Post-Imaginal Growth, and Its Influence on the Taxonomy of Termitophiles

In beetles that live and integrate into termite societies, two well-known morphological adaptations have arisen: the limuloid and physogastric body shapes. Limuloidness presumably evolved as a defensive strategy [23] and is usually associated with a well-sclerotized shield-like body capable of covering retractable and shorter appendages. Physogastry, however, is the enlargement of membranous portions of the abdomen and is much less flexible in terms of adaptation than limuloidness, appearing almost exclusively as a specialization for termitophily, and supposedly functions as mimesis.

Physogastric termitophiles emerge from the pupa as ordinary staphylinids, and the most extravagant modifications seem to occur in imaginal life [24]. Although it is widely known, even in the older literature, it has been shown to mean much more than merely increased body size [4,25]. Hence, post-imaginal growth can often present difficulties for taxonomy, even when the author is aware of the phenomenon, which is immediately acknowledged when we access the literature of some physogastric Termitopaediini, with stenogastric and physogastric specimens being or not recognized as different taxa [4,26,27]. *Nigriphilus mexicanus* (Seevers, 1960) (Corotocini) was initially not acknowledged as a new genus because the author thought the specimens available were intermediate stages of the process of post-imaginal growth of another distinct genus, *Thyreoxenus* (Seevers 1960) [28]; later, the new genus was acknowledged, and Seevers’ specimens were considered fully developed individuals of the then-new genus, *Nigriphilus* [22]. In this context, *Pareburniola dominicana* gen. et sp. Nov. is described based on a unique stenogastric fossil individual. The challenge is to morphologically argue what should and should not change during post-imaginal growth through the last physogastric stage of imaginal life. This is possible, to some extent, by looking at what is already known of the post-imaginal development in other genera, especially those supposed to be closely related to the subtribe Corotocina.

The longitudinal suture on the vertex, although present in the final stage of many Corotocini taxa, has already been shown to be present in the stenogastric stages in *Corotoca* and then to be hidden through secondary sclerotization during post-imaginal growth [7]; thus, this sequence of development may be unknowingly common in the tribe, and its later absence in *Pareburniola* dominicana gen. et sp. nov. is a possibility. In contrast, the size, proportion, and shape of the gula, eyes, and antennae were never recorded to have changed during post-imaginal growth.

The pronotum is usually feebly sclerotized in stenogastric individuals; impressions on the pronotum are usually weakly present in stenogastrics, but there are conditions such as carina and robustness that can only be observed in later forms. For instance, the strong impression on the pronotum in *Eburniola* is unlikely to be present in physogastric *Pareburniola dominicana* gen. et sp. nov., but there is no guarantee that a carina will not appear. First recognized by Seevers [4] for *Thyreoxenus*, legs are known to change during post-imaginal growth; the change is usually about the length of the femur and tibia, which appears to shrink a bit during secondary sclerotization. Tarsomeres, however, are unlikely to change. Elytra do not change, but membranous wings are commonly shed in many groups, which is not surprising given the life habits [29]; *Eburniola* species, however, all have wings in their fully developed forms.

Changes in the sclerites of the abdomen are as expected as the remarkable membranous enlargement of the abdomen. It usually makes the early adult stages (stenogastric) look strikingly different from fully developed ones (physogastric). Some groups go even further in these modifications with pseudoappeandages and overall new shapes for the sternites and tergites, making the sclerites of the abdomen unreliable in the diagnosis of a taxon based on stenogastric individuals. However, setae and chaetotaxy never change, including those in another tagma, and are reliable for diagnosing taxa [7]. Finally, although termitophiles cover only a narrow range of color patterns, varying from light brown to dark brown and black, it is worth noting that stenogastrics can be lighter, with darker pigmentation, if any, appearing later in development.

### 4.2. Biogeography and the Importance of the Fossil for Future Systematic Studies on Corotocini

Seven of the twelve subtribes of Corotocini have genera with species distributed in the Neotropical region, with five being exclusive to this region. However, only a few records exist from Central America, specifically from Costa Rica, Honduras, Panama, and Trinidad and Tobago, all of which are of *Nasutitermes corniger* (Motschulsky, 1855) and/or *Nasutitermes ephratae* (Holmgren, 1910). Within Corotocina, only *Eburniola gastrovittata* Seevers, 1937 (Panama) and *Thyreoxenus parviceps* Mann, 1923 and *T*. *pulchellus* Mann, 1923 (Trinidad and Tobago) are recorded to occur in Central America. *Pareburniola dominicana* gen. et sp. nov. is the first record of Corotocini in the Dominican Republic.

Biogeographic analyses for Corotocini are virtually non-existent, with speculative theories that are often based on vicariance, are unsupported by current evidence, and are mainly from hosts. Corotocina is associated exclusively with Nasutitermitinae, and host data currently support an African origin with an estimated time of about 50 Ma [30]. Even considering the whole of Termitidae, the higher termites are still younger than the event of the Gondwana breakup. Bourguignon et al. [30] argued for five dispersion events to South America in the Oligocene (~30 Ma), which slowed down to just one in the Miocene (23~5.3Ma); hence, it is reasonable to argue that termitophiles reached the Neotropical region along with termites in one or more of these “out of Africa” events. The discovery of *Pareburniola dominicana* gen. et sp. nov. fossil is essential for future studies on the yet poorly understood Corotocini evolution since it should allow temporal inference in dating analyses while also providing a phylogenetic signal for morphology-based phylogenies.

## 5. Conclusions

We described the very first fossil of Corotocini from Miocene Dominican Republic amber. It is included in the subtribe Corotocina, with a possible close relationship with the Neotropical genus *Eburniola*. The representatives of the tribe, and especially from the subtribe Corotocina, undergo profound changes during post-imaginal growth in membranous and sclerotized parts, resulting in a later physogastric stage that is morphologically different from many aspects when compared to the early stenogastric stages. Therefore, the fossil, which has been described as a stenogastric individual, poses a challenge in description. We addressed this by applying current knowledge of how termitophiles undergo post-imaginal growth and what the final stage may or may not look like at the end of that process. This will contribute to future taxonomical works in the tribe, urging for the inclusion of the intermediate stages of development in discussions.

## Figures and Tables

**Figure 1 insects-13-00614-f001:**
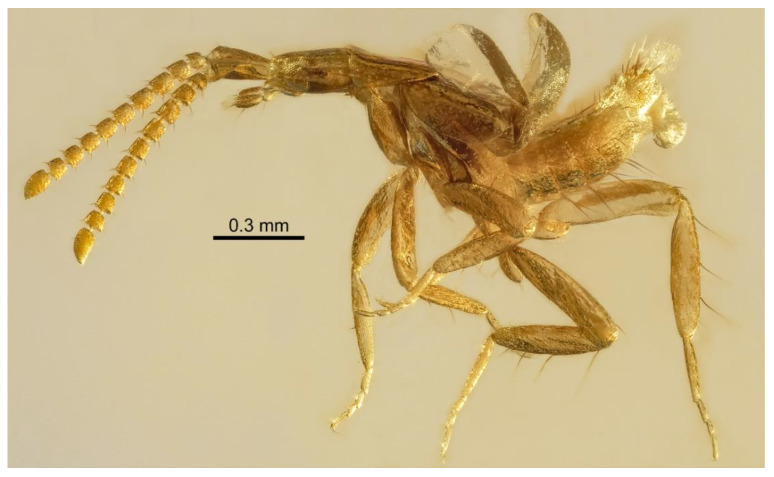
*Pareburniola dominicana* gen. et sp. nov. Habitus, in lateral view. Photo credit: A, P. Gouveia.

**Figure 2 insects-13-00614-f002:**
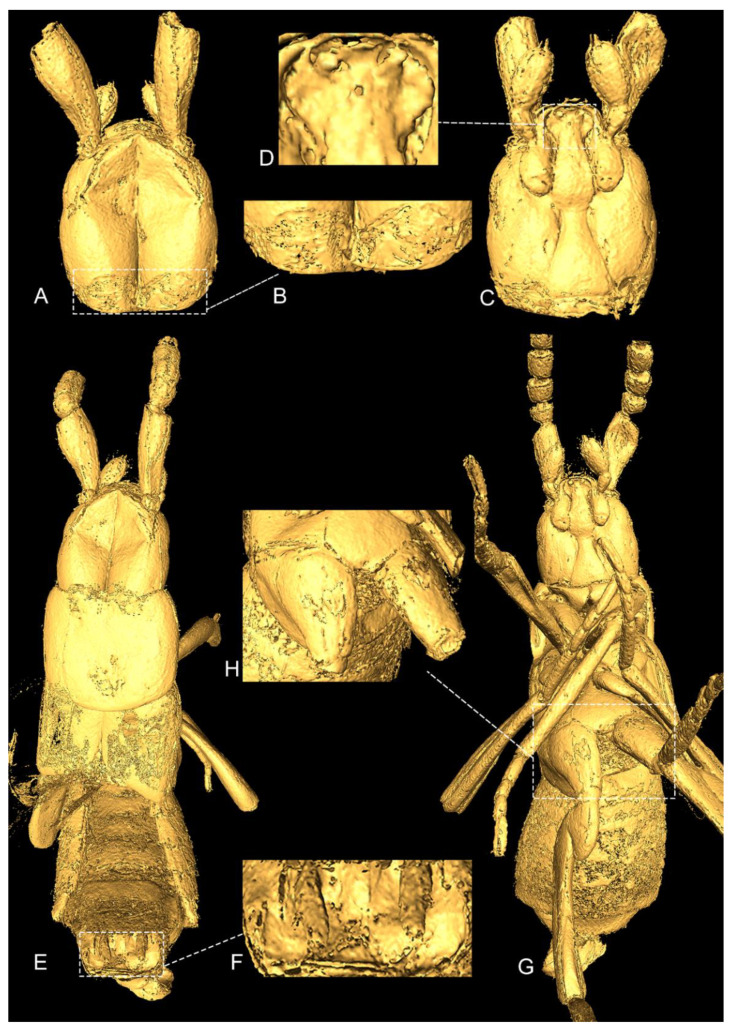
*Pareburniola dominicana* gen. et sp. nov. Head, dorsal (**A**); details of glandular region (**B**); ventral (**C**); prementum (**D**); habitus, dorsal (**E**); segment IX (**F**); ventral (**G**); metasternum (**H**).

**Figure 3 insects-13-00614-f003:**
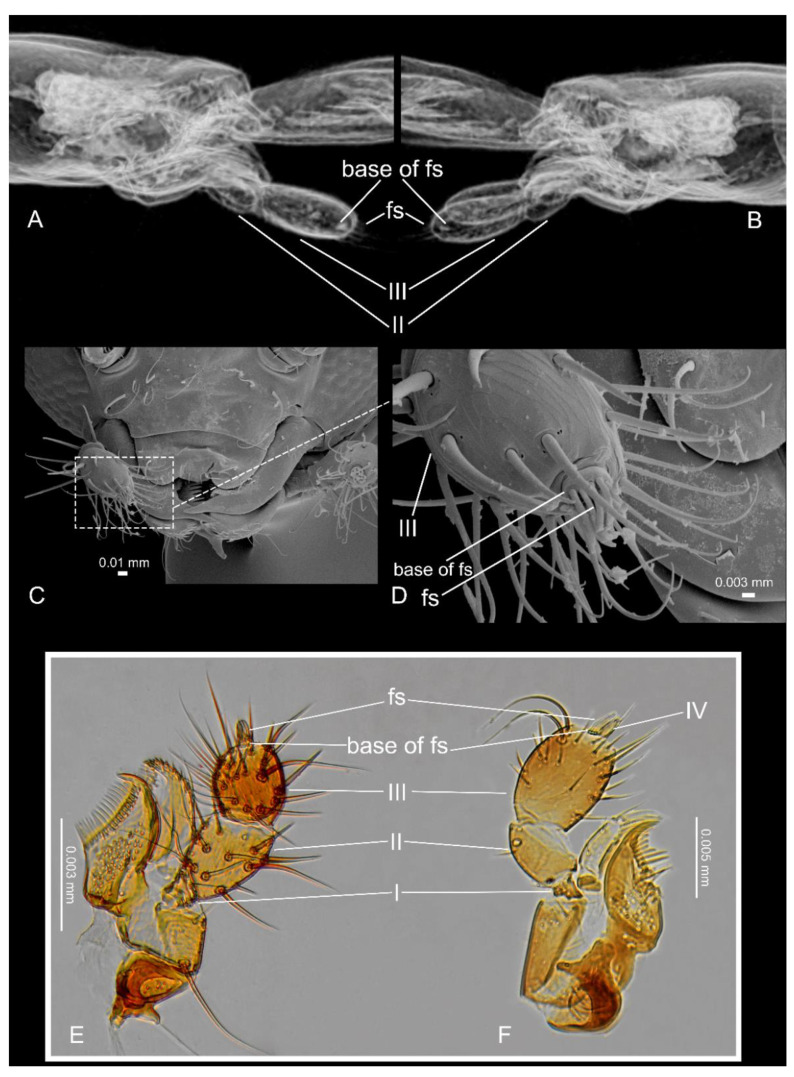
*Pareburniola dominicana* gen. et sp. nov. Head, in lateral view: right (**A**), left (**B**); *Corotoca pseudomelantho* Zilberman, 2020. Head, in frontal view: (**C**), maxillary palp and filamentous sensillae (**D**), whole maxilla (**E**); *Thyreoxenus solomonensis* Seevers, 1937. Maxilla (**F**). fs = filamentous sensillae.

## Data Availability

The data presented in this study are available in the present article.

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
