# Peer review of "The Earliest Corotocini (Insecta: Coleoptera: Staphylinidae) from Dominican Amber, with Remarks on Post-Imaginal Growth Influence on Termitophile Taxonomy†"

_insects, 2022, doi:10.3390/insects13070614_

Round 1

Reviewer 1 Report

No comments to authors

Author Response

No comments from reviewer 1.

Reviewer 2 Report

The authors describe the first fossil representative of tribe Corotocini, which are well-known for their obligate termitophily and strong modifications of the adult body after emergence from the pupa. Overall, this an important discovery for our understanding of the evolution of termite-beetle associations and the emergence of this bizarre type of development. The authors have also produced microCT scans of the fossil and reconstructed images from this 3-D data in standard views. It would be important that the authors deposit the microCT scan data into an open data repository, I did not see any such statement in their manuscript. 

In addition to the taxonomic description of the new genus and species, the paper also includes an interesting discussion on what the final physogastric form of this adult may or may not look like, and the importance of considering this when describing new genera. 

My main criticism of this paper is that it does go into enough morphological  detail with the illustrations (i.e., no labels, most characters described are not shown in illustrations, etc) and the argumentation for placement of the fossil. The authors could have gone further with the microCT data, by segmenting out the legs to show important features such as the mesocoxae (tribe placement) or the tergite X, which they stated as important for the generic diagnosis and relationship with Eburniola. 

For Corotocini: illustrate evidence of coeloconic sensilla; unmargined mesocoxae etc. The antenna looks very well preserved for imaging via light microscopy but the image is too zoomed out

For Corotocina: which morphological definition of this subtribe is being used? A reference would be good here. Based on Jacobson et al. 1986 - there are other groups with 4-4-4 tarsal formula, are other diagnostic features possible to illustrate and label to provide better justification for placement. The metacoxae are stated by Jacobson et al. 1986 to be important for the diagnosis of Corotocina, can these be reconstructed in detail using the microCT data to better justify placement?

For a close relationship with Eburniola, one of the cited characters was the long gula but in the only image of the genus in Jacobson et al. 1986, the head capsule is rather short and therefore, so is the gula. Is this perhaps variable within the genus? The shape of female tergite X seems far more convincing but I cannot see this in the illustrations provided. I also urge the authors to consider non-Neotropical genera of Corotocini, in case they have not. Dominican amber is well-known for including taxa that are currently known only from other biogeographic regions.

Finally, for the genus name, I think 'Pareburniola' would be more correct when combining 'para' and a name beginning in a vowel

With some added justification for fossil placement and for the description, and the deposition of their microCT scan data, I would recommend the publication of this manuscript in Insects

Author Response

We appreciate all the comments and other efforts made by Reviewer 2.

The main criticism from Reviewer 2 is the evidence for the placement of the fossil. In addition, some characters and comparisons should be better explained, as well the addition of more figures. We addressed these issues in the new MS version, including novelty concerning the fourth maxillary palp, strengthening the arguments. 

For Corotocini:

It is sometimes possible to see the coeloconic sensilla. Still, the antenna is treated with KOH most of the time before it. It is worth mentioning that the Corotocini do not have any unique character for the group, any one of each can be found in other groups (for example, Termitonannini also have coeloconic sensilla in the last antennomere). So, for instance, even though it is a Corotocina or Termitoptochina character, the glandular region on the head is definitely better evidence that it is Corotocini than the other character mentioned in the MS.    But in the end, it comes to a convergence of evidence.

For Corotocina:

The tricky thing is separating it from Termitoptochina, as they share 4-4-4 tarsal formula and glandular structure on the posterior region of the head (mostly Corotocina, but it is also present in two genera of Termitoptochina), and the reduction of mouthparts many times. There is also an increase in physogastry, but it really depends on the genus. Also, physogastry is not really useful here as we only have a stenogastric specimen. The lateral extensions of metendesternite is an important character and should have been more emphatic in the text, even though Parerbuniola has the metacoxae just narrowly separately (as in Eburniola)

Another character shared by Corotocina and Termitoptochina, not explicitly mentioned in the literature, is the reduction of the fourth palpomere in some groups. The confusion with this is better explained in response to referee 3. This reduction has never been recorded in other tribes in Corotocini. Termitoptochina also has the labial palps usually more reduced than Corotocina, even though some genera in the latter also might have it reduced, like Thyreoxenus or the extreme case of Austrospirachtha. Parerbuniola has the labial palps developed, even though it was not possible to count the articles. Termitoptochina genera also have a distinct gula, much shorter in length and wide at apex than most Corotocina.

Concerning other groups with tarsal formula 4-4-4, except Termitoptochina:

Indeed there are other taxa with tarsal formula 4-4-4, but they are easily ruled out when we look at some characters. For example, none of them has the fourth labial palp reduced, bear glandular structure on the posterior region of the head, have lateral extensions on the metendosternite, or tergite X divided into two lobes.

For a close relationship with Eburniola:

Yes, it is variable within the genus; actually, it is useful for species-level identification. E. leucogaster and E. lujae (especially E. lujae) have more elongated heads.

We considered non-Neotropical taxa, but it was missing in the text: they lack most of the characters found in Parerburniola. We added further discussion throughout the text.

Reviewer 3 Report

See PDF

Author Response

We appreciate all the comments and other efforts made by Reviewer 3.

The following is the response to Reviewer 3 (lines from the PDF file, not DOC)

Line 15 – successful -> diverse

Line 25 – this secondary development process -> this phenomenon

Trying not to repeat “post-imaginal growth” in the same sentence.  

Line 29 – “seemingly” deleted

It is definitely there; we will highlight it in the figures.

Line 51  – Trichopseniine -> trichopseniine

Line 52 – Yes, but the sentence talks about the Dominican amber, from which the fossil is being described.

Lines 57-58 – The reasons are discussed throughout the work.

Line 89 – abdomen -> abdominal

Line 99 – visible interarticular membrane between them -> pedicels of the antennomeres visible between articles

Line 99-100 – shorter antennomere 10 -> antennomere 10 shorter than other antennomeres except the 3

Line 100 – elytra -> elytron

Line 101 – pro-meso legs -> meso- and meta legs

Lines 122–124 – fourth palpomere, if present, short -> fourth palpomere reduced

Looking at the scans, it seems present, but we just added figures that tell otherwise. There is a high possibility it should be just filamentous sensillae usually present in the tribe (and in Aleocharinae), and the palpomere IV being reduced to a point it is indistinct. It occurs in some Corotocina and Termitoptochina: Corotoca, Eburniola, Termitomimus (also not completely reduced) (Corotocina), and  Lacessiptochus (Termitoptochina), to name a few. When something such as “palpomere IV short” appears in the Corotocini literature, it is probably the basis of the filamentous sensillae mistaken for the maxillary palp.

Furthermore, we are grateful for this observation, as this condition is present in only a few genera, and should be used to help with the taxonomic placement of the fossil. So we just added it into the discussion.

Line 125 –  “seemingly” deleted

Line 128 – elytra -> elytron

Line 128 – wings -> hind wings

Line 130 – separated by a narrow gap between their insertions -> narrowly separated

Line 131 – pro-meta -> meso- and meta

Line 147 – pro-meso -> meso- and meta

Line 155 – mimetic -> mimesis

Lines 156-157 – “Besides any apparent differences between physogastric and limuloid beetles, one has 156 probably been overlooked over the years of termitophilic studies: post-imaginal growth.” deleted

Line 158 – common, usual, customary…

Line 164 – [4,25,26] -> [4, 25, 26]

Line 200 – do not show an extended color gradient, -> cover only a short range of color patterns, varying from light brown to dark brown and black, 

Line 215 – high -> higher

Line 217 – in Oligocene (~30 Ma), slowing down to just one in Miocene -> in the Oligocene (~30 Ma), slowing down to just one in the Miocene

Line 222 – topologies -> phylogenies